# SENTIMENT-ENHANCED STOCK PRICE PREDICTION: A NOVEL ENSEMBLE MODEL APPROACH

## ABSTRACT

Stock price prediction remains a formidable challenge within the realm of financial markets, wherein a multitude of models and methodologies have been under exploration to prognosticate the dynamic behaviour of equities. This research endeavour encompasses an exhaustive examination of extant stock prediction systems, entailing a meticulous assessment of their merits and demerits, concurrently pinpointing discernible lacunae and avenues for enhancement. Subsequently, we harnessed the capabilities of BERT, an exemplar in the domain of natural language processing, to conduct sentiment analysis across a heterogeneous corpus of news articles pertinent to the subject stocks. Additionally, an ancillary sub-experiment was conducted to ascertain the relative impact of three distinct categories of news articles, namely headlines, summaries, and a composite amalgamation of the two, on the efficacy of stock price prediction. The outcome of this investigative pursuit was the generation of sentiment scores for each trading date, which were subsequently integrated as input features in the training of a neural network. Through a comparative analysis of various neural network models, including but not limited to RNN, LSTM, GAN, and WGAN-GP, we discerned that the WGAN-GP model exhibited the most favourable predictive performance. Building upon these findings, we introduced the FB-GAN model, an ensemble architecture comprising WGAN-GP, which capitalizes on the fusion of historical stock price data and market sentiment scores for enhanced stock price prediction. Subsequently, a comprehensive evaluation of our approach was undertaken vis-à-vis established models, gauging its performance against five prominent equities, namely Amazon, Apple, Microsoft, Nvidia, and Adobe. In summation, this research makes a compelling case for the integration of BERT-based sentiment analysis within the ambit of stock price prediction. Our initial hypothesis regarding the significant influence of market sentiment on stock price prediction was validated, and our proposed FB-GAN model outperformed all other models. Furthermore, incorporating both the headline and summary of the news article contributed to enhanced stock price prediction compared to utilizing either the headline or summary in isolation.

## 1 INTRODUCTION

With information transmitting at the speed of light and market dynamics being influenced by an intricate interplay of factors, accurate stock price prediction remains a challenge of paramount importance. Traditional quantitative models, although effective to a certain extent, still struggle to capture the nuances of market sentiment and rely majorly on historical stock price data. In this context, Artificial Intelligence models have played a vital role in predicting stock prices, offering the potential to unravel hidden patterns and provide more accurate forecasts.

The proliferation of social media platforms, financial news outlets, and online forums has revolutionised the accessibility of market-related information. Consequently, investor sentiment, characterized by emotions, opinions, and beliefs, has emerged as a dynamic force capable of triggering rapid shifts in market trends. Based on a study (Liu et al., 2022), the qualitative aspects of investor sentiment profoundly impact market movements, affecting the desired rate of return of the investors. By harnessing the power of natural language processing (NLP), AI systems can parse and comprehend vast amounts of textual data generated daily, and gauge the collective sentiment of market participants.

In this research, we will explore and assess the performance of the already existing Artificial Intelligence models for stock price prediction and come to a consensus on which model provides accurate stock price prediction based only on historical price data, which we will later use to improve and test our hypothesis. The central hypothesis of this research is to incorporate market sentiment data into AI-based stock price prediction models, which can significantly enhance the accuracy and robustness of stock price predictions.

The core research methodology involves designing and implementing an AI-driven framework that seamlessly blends investor sentiment analysis with state-of-the-art predictive algorithms to provide accurate stock price prediction. The predictive accuracy and performance of the proposed model FB-GAN are rigorously evaluated and compared against traditional models using appropriate metrics.

As the financial landscape continues to evolve and adapt to technological advancements, understanding the interplay between AI, sentiment analysis, and stock price prediction becomes imperative. This project not only contributes to the growing body of research in the area of stock price prediction but also enables investors to make more informed decisions based on the proposed model.

## 2 RELATED WORK

In this section, we briefly review renowned techniques presented for stock price prediction. This literature review provides an overview of key developments, methodologies and challenges in the field of stock price prediction, drawing on a range of formative studies.

Time Series Analysis had set the initial groundwork for stock price prediction. Fama (1965) used autoregressive models to capture the serial correlation in stock prices. These methods, however, often assume stationarity and are not capable of capturing the complexities of time series data.

Zhang (2005) used SVMs for short-term load forecasting, which demonstrated the potential of SVMs for time-series prediction tasks such as stock price prediction. Huang et al. (2005) demonstrated the use of SVM with technical indicators for stock price prediction, highlighting SVM's ability to capture complex patterns in financial data.

Heaton et al. (2017) suggested that LSTM architecture can be used as an oscillator, and they concluded that deep neural network can detect patterns in financial data that is hidden even scientists. Liu et al. (2022) developed an LSTM-GRU system to perform short-term stock price prediction. (Chung et al., 2014) mentioned in their work that GRU supplemented LSTM, which accelerated the training and mitigated the problem of overfitting.

Research has been conducted to use sentiment analysis on social media and news data for stock price prediction. Bollen et al. (2011) demonstrated the correlation between Twitter sentiment and stock price movements. Das & Chen (2007) developed an opinion extractor using NLP and sentiment analysis to classify news articles as positive, negative, or neutral, to demonstrate the value of sentiment-based features in predictive models.

Lin et al. (2021) used generative adversarial networks (GANs) to predict stock prices using gated recurrent units (GRU) as a generator that takes historical stock prices as input and generates future stock price, and a convolutional neural network (CNN) as a discriminator to discriminate between the real and the predicted price. Lin et al. (2021) used WGAN-GP to improve and stabilize the training of the GAN model, and thereby improving the stock prediction capabilities.

Devlin et al. (2019) introduced BERT, a transformer-based model, which has created a great impact on NLP tasks. BERT has gained popularity for sentiment analysis, extracting valuable insights from news articles, social media, and financial reports. Akita et al. (2016) demonstrated the effectiveness of BERT-based sentiment analysis in capturing the market sentiment, highlighting the impact of sentiments in improving stock price prediction accuracy.

Sonkiya et al. (2021) examined and compared the performance of different machine learning and neural methods for stock price prediction, including LSTM, GRU, vanilla GAN and Auto-Regressive Integrated Moving Average (ARIMA) model. They build an ensemble model using GAN to predict stock price based on multiple features such as technical indicators, commodities and historical prices with sentiment scores calculated by performing sentiment analysis using on news headlines of Apple Inc.

## 3 METHODOLOGY

This section introduces the data collection, data preprocessing, feature engineering, experimental setup, study of existing stock price prediction models and our proposed model.

### 3.1 DATA COLLECTION AND FEATURE ENGINEERING

#### 3.1.1 DATA COLLECTION

For this study, we selected five stocks: Amazon, Apple, Microsoft, Nvidia and Adobe for stock price prediction based on their 5-year stock price history and market sentiments. The data collection has been done in two phases for this project. In the first phase, we gathered news articles related to a particular stock, and in the second phase, we collected the historical price history of the stocks. The news articles related to a particular stock were collected using the Alpha Vantage API.

Our aim was to conduct this study only with high-quality news articles from trustworthy sources. Alpha Vantage is one of the news aggregators that provides high-quality news articles published by renowned publishers such as Motley Fool, MarketWatch, Bezinga, etc. Another reason to choose Alpha Vantage was that it provides both the headline and a summary of the news articles, which are essential data points we need to study the performance of stock price prediction when market sentiment is calculated based on the type of news article (headline, summary, headline-summary). Although Alpha Vantage provides high-quality news articles, it only has stock news available up to a certain time period. Due to this reason, we could collect stock news information from 01 Mar 2022 to 31 Jul 2023. To handle the missing data, we have taken certain measures, which have been explained in the further section 3.1.2.

Historical price data of the stocks was collected using Yahoo Finance's python package yfinance, which allows users to get data related to a particular stock's close price, open price, high, low and volume for the given time frame. The historical price history collected is from 01 Aug 2018 to 31 Jul 2023.

#### 3.1.2 DATA PREPROCESSING

After performing Exploratory Data Analysis (EDA), we found out there exists some duplicate news articles collected during the data collection of news articles using the Alpha Vantage API. To handle duplicate entries, we used the .duplicates() function of the pandas dataframe to remove the duplicate rows based on headlines and summary columns. We also made use of the relevance score provided by Alpha Vantage, which is a measure of how relevant a news article is to a certain stock. To perform the experiments on the true sentiment score of the headline, summary and headline-summary, we multiplied the relevance score of the particular stock with the three sentiment scores separately. Since Neural networks can only process numerical input, we pre-processed data before feeding it to the network. The two types of inputs we fed the neural network were the stock price data and market sentiment data. The stock price data was already in numerical form; however, the output from FinBERT sentiment classification (refer to section 3.1.3) was in the form of textual labels, namely, positive, negative and neutral. We assigned positive articles a value of 100, negative articles a value of -100 and neutral articles 0. To calculate the sentiment score for a particular day, we define $Sentiment\ Score_n$ as:

$$Sentiment\ Score_n = [\sum_{i=1}^{N}(confidencescore_{pos} * (100)) +$$

$$\sum_{j=1}^{M}(confidencescore_{neg} * (-100)) +$$

$$\sum_{k=1}^{P}(confidencescore_{neu} * (0))]/(N + M + P) \quad (1)$$

where $N$, $M$, and $P$ represent the total number of positive, negative and neutral articles for a particular day, respectively, $confidencescore\_pos$ represents the confidence score of the positive article(s), $confidencescore\_neg$ represents the confidence score of the negative article(s) and $confidencescore\_neu$ represents the confidence score of the neutral article(s).

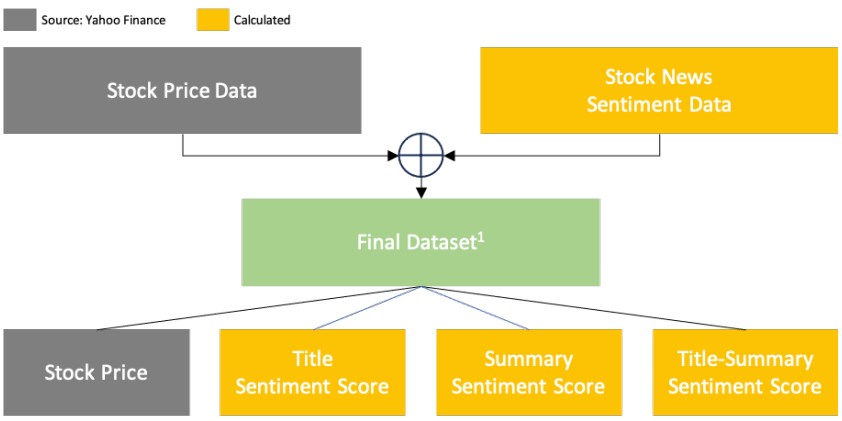

Figure 1: Merging stock price and sentiment score data

To develop the final dataset for the proposed model, we have used the stock price from Yahoo Finance and sentiment scores for each day, using Equation 1, and combined them on the basis of US Trading dates, refer to Figure 1. While combining the stock price and sentiment data, we assumed market sentiment for a particular day would have an effect on the next day's closing price. To handle the dates with no news articles, we have assumed the sentiment for those dates to be neutral i.e., 0 sentiment score.

### 3.1.3 SENTIMENT ANALYSIS USING FINBERT

To categorise a piece of news into a particular category, we performed sentiment analysis on each news article using an AI-based model specialized in Natural Language Processing known as Fin-BERT. FinBERT (Araci, 2019) is a pre-trained BERT (section 3.16.1) model fine-tuned for financial sentiment classification. FinBERT analyses a textual input and provides an output between 0 and 1 and the sentiment label: positive, negative and neutral. A higher score indicates a higher confidence in the label. We pass stock news information of all three types: headline, summary, and headline-summary to the FinBERT model to obtain the category label and the confidence score for the given news article, as shown in 2.

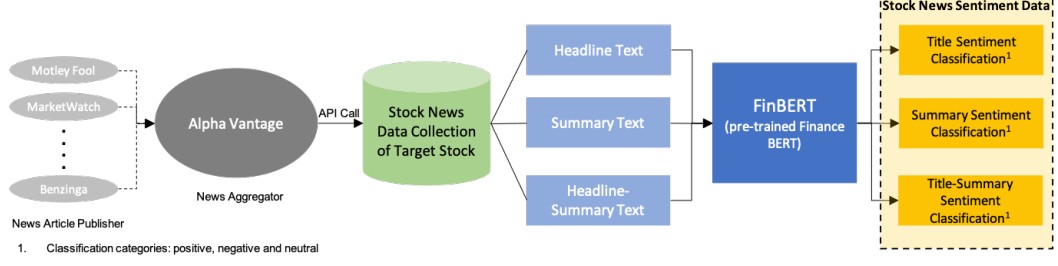

Figure 2: Sentiment Analysis Diagram Flow

### 3.1.4 EXPERIMENTAL SETUP

To conduct the experiments, we have used Python 3 Google Compute Engine. The hardware setup includes a Nvidia Tesla T4 GPU with a 2-core Intel Xenon CPU 2.2 GHz, supported by 13GB RAM and 16 GB GPU Memory.

## 3.2 EXISTING MODELS

Our approach to developing the proposed model has been based on selecting the most accurate stock price prediction model and then modifying the model to leverage the market sentiments to predict the stock price. In the following sections, we have discussed the pre-existing models such as Vanilla RNN 3.2.1, LSTM 3.2.2, GAN 3.2.3, and WGAN-GP 3.2.4. 80% of the samples are used for training, and 20% are used for testing the models.

### 3.2.1 VANILLA RNN

Recurrent Neural Networks (RNN), are designed in such a way that they are able to generate output values leveraging information from previous trends. In this model's experiment, we considered the stock price of 3 days (3 data points) to forecast the prices for the next day (1 data point). This data structure is called many-to-one. The input is a 3D array having 3 dimensions: tensor, time step and feature. Tensor is a vector that enters the model, the time step is one observation in the tensor, and the feature is one observation at a time step. The predictions are done based on the Adj. Close price as the input feature. The Simple Recurrent Neural Network is implemented based on the Keras, deep learning framework, with TensorFlow backend. This simple RNN model has 5 layers: 1 input layer, 3 hidden layers, and 1 output layer. The optimizer used was Root Mean Square Propagation (RMSprop), the loss function was Mean Squared Error (MSE), the number of epochs was 100, and the batch size was 150.

### 3.2.2 LONG SHORT-TERM MEMORY

As discussed in section 2, LSTM are an improved version of RNN which can retain information over longer periods and help solve the problem of vanishing or exploding gradient. In the LSTM model, we used a similar input vector as we did in the case of simple RNN (section , where we used 3 days of Adj. close price to predict the Adj. close price of the next day. This LSTM model is also implemented based on the Keras deep learning framework. The LSTM model has 5 layers: 1 input layer, 3 hidden layers, and 1 output layer. The optimizer is Adam, the loss function is Mean Squared Error (MSE), the number of epochs is 100, and the batch size is 150. To prevent model overfitting and improve model performance, a dropout layer is added after each LSTM layer, and the dropout ratio is set to 0.2.

### 3.2.3 GAN

The concept of Generative Adversarial Networks (GAN) introduced by Goodfellow (2016) was originally developed to generate synthetic images. However, based on the work done by Lin et al. (2021), GANs also be used for stock price prediction. GRUs were used as the generator based on their robustness and stability, and Convolutional Neural Networks (CNN) were used as the discriminator to distinguish whether the input data was real or fake. The input for the generator was a three-dimensional array of tensors, time steps and features, similar to vanilla RNN (refer to section 3.2.1). However, the model GAN was trained on 6 features: Adj. Close, Open, High, Low, Close and Volume, using 3-time steps to give the prediction of the next day's Adj. close price. The optimizer used was Adam, with a learning rate of 0.00016, the number of epochs was 165, and a batch size of 128. Leaky Rectified Linear Unit (ReLU) is used as an activation function among all layers except the output layer, which is a sigmoid activation function. The sigmoid function basically gives a single scalar output of 0 and 1, which represents real or fake, and the linear function gives a scalar score. The model is tuned with a learning rate between 0.0003, number of epochs of 300 and a batch size between 64 to 512.

### 3.2.4 WGAN-GP

The discriminator of the Basic GAN was not powerful enough, and the training process was slow and unstable. WGAN was proposed by Arjovsky et al. (2017) to help stabilize and improve the training of GAN. The architecture of WGAN-GP is almost the same as that of GAN (refer to section 3.2.3); however, the output layer of the discriminator of the WGAN-GP is a linear activation function instead of a sigmoid function, and an additional gradient penalty is added to the discriminator. The benefit of not having a sigmoid function, we get a scalar score as an output instead of probability. The scores are interpreted the same as the real input data (Zhou et al., 2018). The optimizer used is Adam, with a learning rate of 0.000115, the number of epochs is 100, and a batch size of 128. The discriminator and generator are the same as basic GAN; however, the discriminator is trained once, and the generator is trained thrice.

### 3.3 PROPOSED MODEL: FB-GAN

Based on existing research (Lin et al., 2021), WGAN-GP proves to be a better model for stock price predictions than other traditional models. Using this information, we have proposed a model, FB-GAN, a modified version of WGAN-GP that incorporates market sentiment generated by Fin-BERT for stock price prediction. Similar to WGAN-GP, FB-GAN also has two major components discriminator and generator. The discriminator was made up of three 1-dimensional Convolutional Neural Networks having 32, 64 and 128 neurons in the three layers, respectively, with a flatten layer followed by three dense layers and, finally, the output layer, which used linear activation function. The generator was made up of three GRU Units having 1024, 512 and 256 neurons in three layers, respectively; each layer has a dropout ratio of 0.2, followed by three dense layers. The proposed model, FB-GAN, can be visualized in Figure 3. Our proposed model, FB-GAN, is trained on 7 features: Adj. Close, High, Low, Close, Open, and Market Sentiment Score. The market sentiments were fed to the neural network as a latent input (co-variant) along with other inputs. As we already

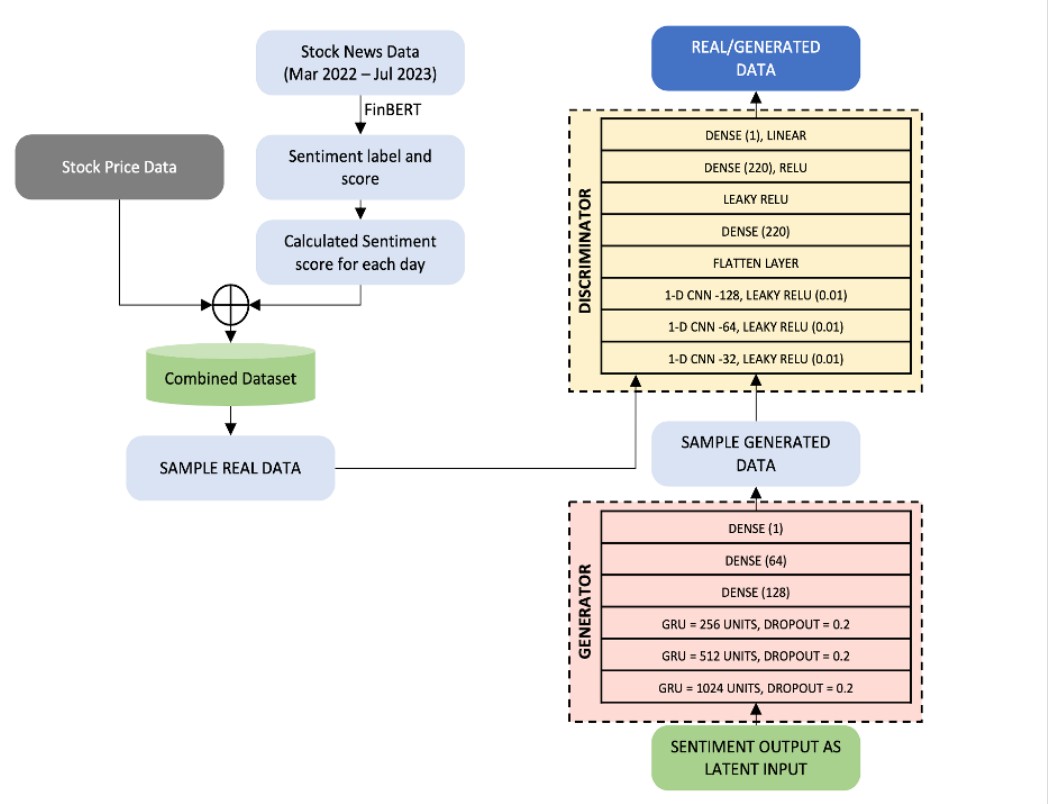

Figure 3: Proposed Model: FB-GAN

mentioned, we have an objective to study the performance of the type of market sentiment information on the stock price prediction. We used the three market sentiment scores calculated using headline, summary and headline-summary for performing the experiments. The optimizer used was Adam, with a learning rate of 0.000128, the number of epochs is 160, and a batch size of 128. 80% of the samples are used for training, and 20% are used for testing the model.

## 4 RESULTS AND DISCUSSIONS

### 4.1 QUANTITATIVE ANALYSIS

Along with our proposed model (FB-GAN), Vanilla RNN, LSTM, GAN, and WGAN-GP were also evaluated for comparison, refer to 4.1. For evaluating the model for comparison we have used Root Mean Square Error (RMSE) 2:

$$\text{RMSE}(y, \hat{y}) = \sqrt{\frac{\sum_{i=0}^{N-1}(y_i - \hat{y}_i)^2}{N}} \qquad (2)$$

where $y_i$ is the actual (true) value of the $i^{th}$ data point, $\hat{y}_i$ is the predicted value of the $i^{th}$ data point and $N$ is the total number of data points.

Table 1: Comparison of results of different models based on RMSE

|  | Amazon | Apple | Microsoft | Nvidia | Adobe | Average |
|---|---|---|---|---|---|---|
| Vanilla RNN | 5.30 | 9.34 | 16.24 | 25.98 | 16.66 | 14.71 |
| LSTM | 4.31 | 6.53 | 9.44 | 28.19 | 15.73 | 12.84 |
| GAN | 4.49 | 12.73 | 16.74 | 23.02 | 15.76 | 14.55 |
| WGAN-GP | 4.01 | 4.35 | 18.29 | 18.30 | 14.67 | 11.92 |
| FB-GAN (Headline) | 4.78 | 7.61 | 12.53 | 15.58 | 21.10 | 12.32 |
| FB-GAN (Summary) | 4.30 | 8.13 | 12.26 | 19.01 | 21.67 | 13.07 |
| FB-GAN (Headline-Summary) | 5.03 | 6.98 | 10.08 | 14.19 | 17.76 | 10.81 |
| Average | 4.52 | 7.49 | 13.00 | 20.21 | 17.21 | – |

Root Mean Square Error measures the root mean squared error between the actual stock and predicted stock price value. A lower RMSE value signifies a better model as the predicted values are as close as possible to the predicted values. Our proposed model, FB-GAN, accepts the market sentiment score as a latent input to predict the stock price for a given stock. To understand the effect of stock price prediction with and without sentiment information, we compared our proposed model, FB-GAN (Headline-Summary), with WGAN-GP, and it performed better than WGAN-GP by 9%. Also, our proposed model outperformed all the other models as well.

We conducted the experiment on three different versions of the news articles: headline, summary and headline, using our proposed model, and we obtained the test results as mentioned in Table 4.1. On comparing RMSE values, FB-GAN gave the best results based on the headline and summary combined sentiment for each stock with an average RMSE of 10.81, followed by the headline sentiment with an average RMSE of 12.32 and lastly using the summary sentiment with an average RMSE of 13.07.

Figure 4, showcases the actual vs. predicted stock price of the five stocks under study, obtained through the FB-GAN (Headline-Summary) model using the Test dataset of each stock. Although each of the stocks has complex time series data, the FB-GAN model performs well in predicting the stock price of each stock. From Table 4.1, we can observe that on comparing the average RMSE score of each stock, Amazon has the lowest average RMSE score of 4.52, and Nvidia had the highest average RMSE score of 20.21 based on all the models. This implies stock price prediction for each stock is different and cannot have a similar or near RMSE score.

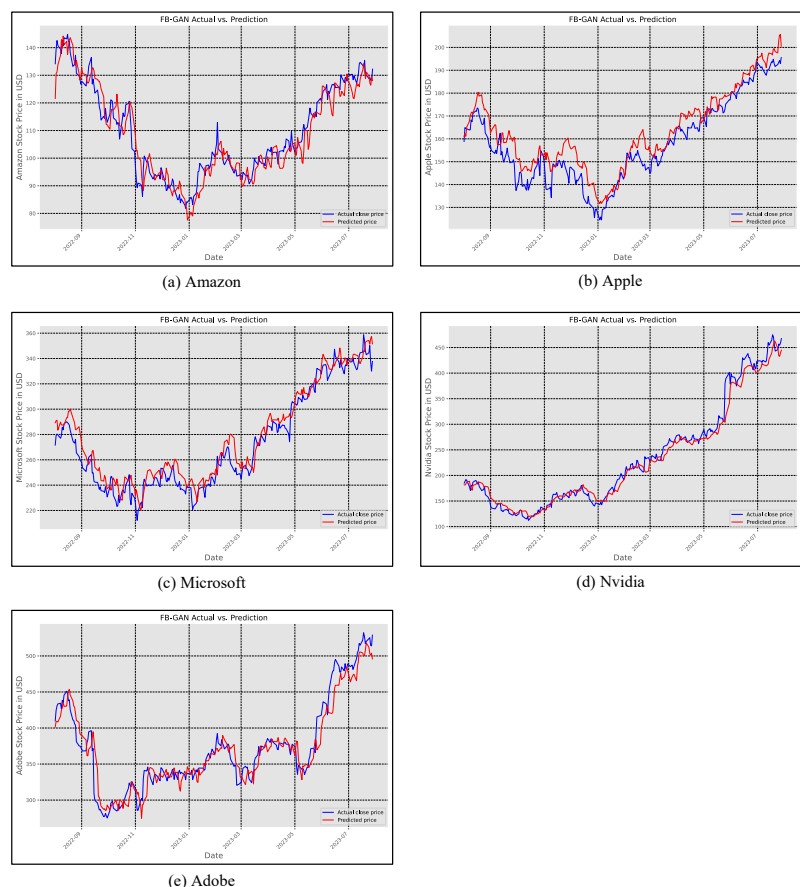

Figure 4: Actual vs. FB-GAN (Headline-Summary) Predicted Stock Price Graphs of different stocks

## 4.2 QUALITATIVE ANALYSIS

In section 3.1.3, we discussed how we performed sentiment analysis using FinBERT on the news articles we had collected using the Alpha Vantage API. We had saved the results which were obtained from the sentiment analysis in the csv format for further experiment process, refer to Figure 5. In the snapshot (Figure 5), column D, E, F named as title, summary and title_summary they represent the headline, summary and headline-summary.

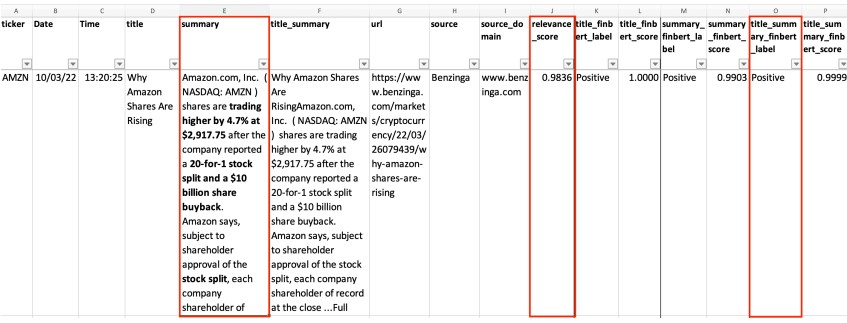

Figure 5: Sentiment classification contrast based on Headline, Summary and Headline-Summary

On performing the back-testing, we compared the results from the FinBERT classification, refer to Figure 5, and the ground truth i.e. actual stock price movement, refer to Figure 6, we observed that the news article mentioning about - " Amazon [. . . ] shares [. . . ] trading higher by 4.7% at $2917.75 after the company reported 20-for-1 stock split and a $10 billion share buyback [. . . ]," was classified as positive with a confidence score of 99.9% and relevance score of 98.4% and when compared with the actual stock price movement after this big news there has been observed an upward stock price movement. Based on the back-testing, we have double verified our hypothesis, that there exist a correlation in market sentiment and stock price movement.

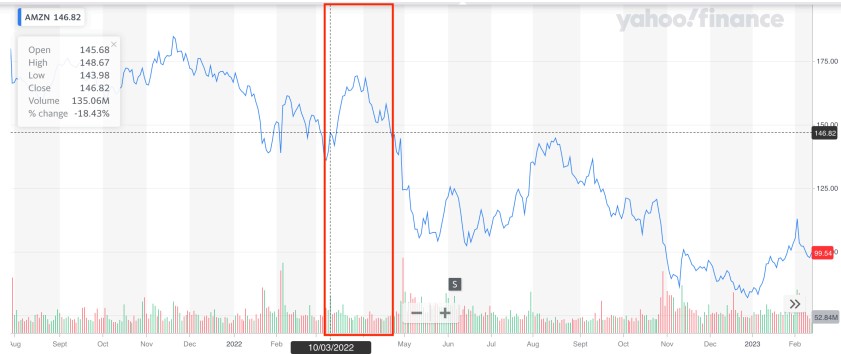

Figure 6: Actual stock price of Amazon with positive price movement highlighted in red box

## 5    Conclusion and Future Work

In conclusion, we can say that our hypothesis, that there exists a strong correlation between the market sentiments on the stock price movement, has been validated. Through this research, we found out that headline-summary combined provide better stock price predictions than headline or summary alone by 12.3% and 17.3%, respectively, based on average RMSE scores. Also, our model outperformed all the other existing models.

As stated in the EMH (Efficient Market Hypothesis), stock prices reflect all the past information, including historical or current stock prices, market sentiment, insider information, financial statements, economic data etc. Based on the EMH theory, our proposed model can be improved if more correlated factors, such as gold prices, bank rates, etc., are leveraged while training the model for stock price prediction. Another future aspect of work involves modifying our proposed model to take into account the real-time stock price and market sentiment data to predict the stock prices which can be used for Intra-day trading. Lastly, based on the trends in the field of machine learning, Transformers can also explored for stock price prediction.

### Author Contributions

The major contributions of this paper are summarized below:

- We conducted an extensive comparative study, evaluating the performance of the most widely used machine learning models for stock price prediction, highlighting their strengths and weaknesses.

- We developed a robust framework for incorporating market sentiments into AI-based stock prediction, enhancing stock price prediction accuracy.

- We performed a nuanced sentiment analysis on the headline, summary, and combined headline-summary approach, assessing their impact on prediction accuracy and concluded that headline-summary combined offer better stock price prediction that headline or summary alone.

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
