# OpenReview forum: "Sentiment-Enhanced Stock Price Prediction: A Novel Ensemble Model Approach"
_ICLR.cc/2024/Conference — Submitted to ICLR 2024_

### Official Review · Reviewer_p754 · 2023-10-31

**Soundness:** 3 good
**Presentation:** 3 good
**Contribution:** 2 fair
**Rating:** 3
**Confidence:** 4

**Summary:**

This paper presents FB-GAN, a stock price prediction model based on sentiment analysis. The authors tested the model on historical price data for 5 stocks, and found FB-GAN to outperform a number of baseline models in average. My main concern about this paper is its novelty. The proposed FB-GAN model appears to be a simple extension of WGAN-GP, and the intuition behind the extra modules is not well justified. In addition, the stock universe is too small, which makes the reported results unreliable. I would encourage the authors to explain in more details the design choice of FB-GAN, and to expand the dataset.

**Strengths:**

- This paper presents an extensive study of a number of methods on their performance in stock price prediction.
- The effect of sentiment analysis to stock price prediction is validated in the paper.
- The paper is well-written and easy to follow.

**Weaknesses:**

- The stock universe in the experiments consists of only 5 stocks, which raises concern about whether the proposed approach would generalize to other stocks.
- FB-GAN appears to be a naive extension of WGAN-GP. The authors fail to explain the intuition behind the module designs.
- FB-GAN is worse than WGAN-GP on 60% of the stocks tested.

**Questions:**

- Have the authors tested WGAN-GP with sentiment scores as the input features?
- Have the authors studied the relative improvement after incorporating sentiment scores. An ablation study would let interested readers know the effectiveness of sentiment analysis in stock price prediction.

---

### Official Review · Reviewer_vbhJ · 2023-10-31

**Soundness:** 1 poor
**Presentation:** 1 poor
**Contribution:** 1 poor
**Rating:** 1
**Confidence:** 5

**Summary:**

The authors tried to enhance the accuracy of stock price prediction via news sentiment.

**Strengths:**

N/A

**Weaknesses:**

1. Lack of technical contribution.
2. The overall presentation is in a bad shape.
3. Weak experiments.

**Questions:**

This paper lacks a significant technical contribution, and the overall representation is unclear. I don't think it's a good fit for ICLR.

---

### Official Review · Reviewer_J1EJ · 2023-11-03

**Soundness:** 2 fair
**Presentation:** 3 good
**Contribution:** 1 poor
**Rating:** 3
**Confidence:** 4

**Summary:**

The paper studies the incorporation of sentiment analysis into stock prediction. The paper shows the RMSE between the actual stock price and the predicted stock price is lowered when the sentiment score is added on top of the previous stock price.

**Strengths:**

The paper presents the idea of leveraging sentiments from news articles for stock price prediction. Every component of the model is explained and the readers can easily understand the structure of the model.

**Weaknesses:**

1. The science/engineering contribution of paper is very limited. The paper did not propose new methods of obtaining stock-related news articles, new approaches of de-noising the news articles, new models that are developed to study sentiment-score-augmented stock price prediction etc. The paper gave the reviewer an impression that it is a concatenation of the existing models and data without much deep analysis.

2. The paper is lack of scientific rigor.
* There is no guardrail for the accuracy of the sentiment analysis. That means we don't know the accuracy of the sentiment scores that are fed into the prediction model.
* The authors claimed in the model that "a strong correlation between the market sentiments on the stock price movement". But there is no  statistical analysis to back the statement up.
* We need variable control in the comparison between WGAN-GP and FB-GAN. For example, the model WGAN-GP should have the same input dimension as FB-GAN with the sentiment score set to neural.

3. RMSE is not a good metric overall. It does not translate well to trading strategies because large RMSE does not necessarily mean losing money.

**Questions:**

No further questions.

---

### Meta-Review · Area_Chair_utG7 · 2023-12-10

**Metareview:**

The paper studies how integrating sentiment analysis into stock prediction impacts accuracy. The authors provide a pipeline that integrate price data with news sentiment data to improve predictions and reduce RMSE. The reviewers unanimously agree that the paper should be rejected. At a high-level there are two issues: The primary issue is that the paper lacks significant methodological novelty and it is likely that more sophisticated versions of proposed methods already exist. Secondly, the experiments and evaluations are limited. For instance, evals are on 5 ticker symbols (Amazon, Apple, Microsoft, Nvidia and Adobe) which is not really acceptable.

**Justification For Why Not Higher Score:**

N/A

**Justification For Why Not Lower Score:**

N/A

---

### Decision · Program_Chairs · 2024-01-16

Reject